# Eligibility and GDMT up-titration success in heart failure: A real-world assessment

Annika Weidenhammer[1], Suriya Prausmüller[1], Marc Stadler[1], Noel Panagiotides[1], Georg Spinka[1], Gregor Heitzinger[1], Henrike Arfsten[1], Guido Strunk[2], Anush Barkhudaryan[1,3,4], Clemens Partsch[1], Philipp Bartko[1], Georg Goliasch[1], Christian Hengstenberg[1], Martin Hülsmann[1]*, Noemi Pavo[1]

1 Department of Internal Medicine II, Clinical Division of Cardiology, Medical University of Vienna, Vienna, Austria, 2 Complexity Research, Schönbrunner Straße 32, Vienna, Austria, 3 Department of Cardiology, Clinic of General and Invasive Cardiology, University Clinical Hospital № 1, Yerevan State Medical University, 4 "Yerevan„ Scientific Medical Center, Yerevan, Armenia

* martin.huelsmann@meduniwien.ac.at

## Abstract

### Background

Implementation of GDMT in HFrEF remains incomplete. Prescription manners may vary based on the development of adverse effects. An HFA position paper proposed patient profiling and individualized prescription manners. This study aims to assess the eligibility for GDMT up-titration and its success in the context of clinical profiles in chronic severe HFrEF outpatients.

### Methods

Clinical characteristics of 900 HFrEF patients at first presentation were assessed, and GDMT up-titration limiting factors were identified by applying thresholds of mutual consent. GDMT prescription was analyzed at 6 months and 1 year.

### Results

75% of patients had no GDMT up-titration limiting factor at baseline. Significant up-titration could be achieved in all four HF drug classes, especially within the first 6 months, irrespective of GDMT up-titration limiting factors ($p \leq 0.035$ for administration and dosage, all drug classes). During up-titration, there was a balanced transition between up-titration limiting factors. 35% of patients received triple therapy on target dosages with a 2.6% one-year mortality rate. Regarding the HFA profiles, 62% of patients could not be classified into a specific HFA phenotype, including most severe patients. 98% of classifiable patients belonged to only four phenotypes, while GDMT up-titration could be achieved in all of these HFA profiles ($p \leq 0.007$).

**Data availability statement:** Data cannot be shared publicly because of restrictions based on the informed consent. Data are available from the Office of Data Management, Department of Cardiology, Medical University of Vienna (Email: cornelia.gabler@meduniwien. ac.at) for researchers who meet the criteria for access to this confidential data.

**Funding:** The author(s) received no specific funding for this work.

**Competing interests:** The authors have declared that no competing interests exist.

**Abbreviations:** ACEi: Angiotensin-Converting Enzyme Inhibitor; ARB: Angiotensin Receptor Blocker; BB: Beta Blocker; BP: Blood Pressure; eGFR: Estimated Glomerular Filtration Rate; GDMT: Guideline-Directed Medical Therapy; HFA:Heart Failure Association; HFrEF: Heart Failure with reduced Ejection Fraction; HK: Hyperkalemia; IQR: Interquartile Range; LVEF: Left Ventricular Ejection Fraction; MRA: Mineralocorticoid Receptor Antagonist; NtproBNP: N-Terminal Pro-B-Type Natriuretic Peptide; NYHA: New York Heart Association; RASi: Renin-Angiotensin System Inhibitor; RCT: Randomized Controlled Trial; RHR: Resting Heart Rate; SGLT2i: Sodium-Glucose coTransporter-2 Inhibitor.

## Conclusion

In this real-world study, 75% of patients with chronic severe HFrEF are eligible for GDMT up-titration towards target dosages. The clinical profile concept of the HFA might be adapted, as most classifiable patients can be up-titrated, and most severe patients are missed by the classification.

## Introduction

Guidelines for managing patients with Heart Failure with reduced ejection fraction (HFrEF) emphasize the use of all class I recommended drugs, i.e., Guideline Directed Medical Therapy (GDMT), with early treatment initiation and up-titration to maximum dosages. However, the up-titration of GDMT in everyday practice is frequently delayed and inconsistent [1,2]. In the CHAMP-HF registry, only 1.1% of patients received class I Heart Failure (HF) drugs at the target dosage [1]. This treatment inertia contributes to the persistently high mortality and morbidity rates observed in HFrEF [3–9].

Among other reasons, distinct clinical features, such as arterial hypotension, bradycardia, or impaired renal function with or without electrolyte disturbance are considered to be most responsible for non-prescription or insufficient up-titration of HF drugs. These features are common in HFrEF and abundant in high-risk patients. In the presence of these features, clinicians tend to restrict HF drug use or maintain lower-than-target dosages. However, reasons for refraining from up-titration are not precisely defined and may be subject to personal bias. Implementation trials like STRONG-HF, use cutoffs for up-titration of medication, and these present challenges to provide adequate therapy. STRONG-HF has proven that rapidly sequenced up-titration of HF drugs is feasible and safe in patients hospitalized for acute HF. However, here, patients with an estimated glomerular filtration rate (eGFR) < 30 ml/min/1.73m$^2$, systolic blood pressure (BP) < 100 mmHg, heart rate < 60 bpm, serum potassium > 5.0 mEq/L and a documented intolerance to high doses of beta-blockers (BB) and Renin-Angiotensin System Inhibitors (RASi) (Angiotensin-converting enzyme inhibitors (ACEi) and Angiotensin receptor blockers (ARB)) were excluded [10]. These predefined cutoffs are essential in trial settings but may pose challenges in real-world implementation, where more flexibility is often required to optimize therapy. Other data on consecutive hospitalized HF patients show that 72% are eligible for up-titration of all GDMT classes [11]. In detail, RASi, BB, Mineralocorticoid Receptor Antagonists (MRA) and Sodium-Glucose Transport Protein 2 Inhibitors (SGLT2i) were eligible in 98.4%, 93.1%, 100% and 75.4% of cases, respectively [11,12]. Recently, Greene et al. have shown that 82% of 33.036 patients of the Get With The Guidelines Heart Failure registry hospitalized with de-novo HF qualify for quadruple therapy, yet only a minority receives it, failing to achieve maximum clinical benefit [13]. This study has used different cut-offs for bradycardia (resting heart rate (RHR) ≤50bpm), systolic blood

pressure (≥85–100mmHg), renal function (eGFR ≤ 30ml/min/1.73m2) and potassium levels (≤ 5.0 or 5.4 mmol/l) as a demonstration for the arbitrariness of what clinicians consider as a contraindication. The different cut-offs mentioned here refer to the thresholds applied in the large RCTs that tested and led to the introduction of specific HF medications into clinical practice, which often vary. In addition, the same author described that in outpatients with stable chronic HFrEF within the CHAMP-HF registry, only a minority of patients received all classes of GDMT at baseline, and only <1% were on target dose of triple therapy, even after 12 months [14,15].

A position paper by the Heart Failure Association (HFA) [16] proposed a more personalized approach towards rapid up-titration to target dosages based on a patient's predisposition to develop potential adverse effects. According to the HFA concept, HFrEF patients are divided into 9 profiles characterized by clinical features, which assumedly influence the initiation and up-titration of class I HF drugs. Each profile has individualized therapy recommendations.[16] The feasibility and clinical benefit of classifying the HFA phenotypes have sparsely been investigated in clinical practice [17,18]. Data from the CHECK-HF registry and Swede-HF described the distribution of phenotypes in real-world cohorts. However, it remains unknown how these proposed phenotypes affect up-titration strategies [17,18].

It is important to note that the guidelines recommend consulting a specialized heart failure center instead of adjusting medication, even when faced with conditions like arterial hypotension, bradycardia, or impaired renal function with or without electrolyte disorders [2].

This study aimed to investigate the proportion of patients eligible for GDMT up-titration and up-titration success in outpatients with chronic severe HFrEF after referral to a specialized HF center.

## Methods

The study was approved by the Institutional Review Board of the Medical University of Vienna and conducted in accordance with the current revision of the Helsinki Declaration. The authors confirm that patient consent forms have been obtained in written form for this article. The data used in this retrospective study were accessed on December 6th, 2022. While the authors had access to identifiable information during data collection, these were securely stored and handled in accordance with institutional and ethical guidelines to protect participant confidentiality. No identifying information was included in the analysis or publication.

### Patient population and follow-up time points

The tertiary care outpatient HFrEF clinic of the Medical University of Vienna is an all-comer unit for patients with a recent history of chronic severe systolic HF (defined by a left ventricular ejection fraction (LVEF) <35% and N-terminal B-type brain natriuretic peptide (Nt-proBNP) >1500pg/ml). Patients are referred to the unit to establish GDMT and further evaluate the patients towards advanced therapies, such as heart transplantation. Patients are consecutively included in a prospective registry. A comprehensive set of variables is assessed at first presentation and routine clinical follow-ups. The data includes patients' history, demographics, parameters of cardiac imaging, and medication, including dosages and laboratory values. The MDRD-IDMS calculates sex-adjusted eGFR values. Mortality data are obtained from the database of the official national registry Statistic Austria.

For this study, patients presenting from 2016 to 2022 to the HF unit and a complete data set at baseline were selected. Information on systolic BP, RHR, renal function, serum potassium levels, and the presence or absence of atrial fibrillation were the requirements. Two follow-up time points at 6 months ± 3 months (= 6 months) and 12 months ± 3 months (= 1 year) after baseline were defined to evaluate the development of clinical features and medical prescriptions.

For the evaluation of the HFA concept, patients were categorized at baseline into the different clinical profiles of the HFA position paper [16]. The categorization and adjustments are described in detail in the supplementary material (Supplementary Methods 1 in S1 File).

## Clinical features at baseline and follow-up: the presence of GDMT up-titration limiting factors

The clinical features commonly preventing up-titration were assessed for each visit. Similarly to previous investigations [10–12], we have defined cut-offs in line with the convention for each clinical feature, which is termed an up-titration limiting factor. These were defined as an RHR ≤ 55 bpm while not on BB therapy at target dosages, a systolic BP ≤ 90mmHg while not on RASi therapy at target dosages, an eGFR ≤ 30ml/min/1.73m$^2$ while not on RASi/MRA therapy at target dosages, and the occurrence of hyperkalemia (HK, i.e., potassium ≥ 5.5mmol/l) while not on RASi/MRA therapy at target dosages or receiving treatment with a potassium binder.

## Assessment of HF medication

The use and exact dosages of medications, including class I HF drugs, i.e., BB, RASi (ACEi/ARB/ARNi), MRA, and SGLT2i, are documented within the registry at each visit. The dosages are expressed as a percentage of the target dosage of the respective substance. If patients did not receive a particular drug, the dosage was set at 0%. This study was mainly conducted before the era of the recommendations for SGLT2i. Therefore, the interpretability of the analysis of this drug class is constrained. For simplification, the up-titration success of GDMT was defined by the use and maximum recommended dosages of the three drug classes, including BB, RASi/ARNi, and MRA.

## Statistical analysis

Continuous data are expressed as the median and interquartile range (IQR) and categorical variables as percentages. The presence of GDMT up-titration limiting factors at each time point was assessed. The distribution of up-titration limiting factors within the total cohort at baseline was visualized as a donut chart. The survival of patients with and without up-titration limiting factors at baseline was visualized using Kaplan-Meier curves, and a comparison was performed using the log-rank test between all groups. Baseline characteristics of all GDMT up-titration limiting factors were compared using the Kruskal-Wallis test and the Fishers' Exact test. Implementation and up-titration of HF medications were visualized as histograms for each substance and timepoint according to the clinical profile assigned to them by having GDMT up-titration limiting factors or not. Regarding the intensity of HF medication, patients were grouped into receiving or not receiving specific medication and the percentage of target dosage according to current guidelines [2]. The Wilcoxon (matched pair signed rank test) was used to compare the different time points. The Box-plot: GDMT at 6 months was calculated from all patients with data at 6 months. The longitudinal evolution, i.e., development and disappearance of up-titration limiting factors and transition between groups, was analyzed for a subgroup of patients with a complete dataset at baseline, 6-months, and 1-year follow-up (n = 264). Patients were categorized into the suggested HFA profiles according to S1 Table in S1 File, and the distribution between the different groups was presented. Continuous variables are given as median and 25th and 75th percentile; counts are given as numbers and percentages. Comparison between the group "no GDMT up-titration limiting factors" and patients with GDMT up-titration limiting factors, i.e., the RHR-, SBP-, eGFR- and HK-group, was performed by the Kruskal-Wallis and the Fishers´ exact, the p-value is indicated. A significant p-value is indicated in bold. If our dataset was not complete, an individual n is given, and % is given from the total cohort. The association between HFA profiles and outcomes was analyzed using Kaplan-Meier curves and the log-rank test to compare the groups. Two-tailed analyses were used for all tests, and the significance level was set at $p < 0.05$. Statistical analysis was performed using SPSS®software 26.0 for OS Windows 10.

## Results

### Baseline characteristics of the study population

A total of 900 consecutive patients were included. Baseline characteristics of the overall cohort are shown in Table 1.

**Table1. Baseline characteristics of HFrEF patients grouped according to the presence or absence of factors limiting GDMT up-titration.**

| | No factors limiting up-titration=677 | HR-group n=45 | p-value | SBP-group n=25 | p-value | eGFR-group n=67 | p-value | HK-group n=45 | p-value | multiple factors limiting up-titration n=41 | p-value | Total cohort n=900 |
|---|---|---|---|---|---|---|---|---|---|---|---|---|
| *Basic demographics* | | | | | | | | | | | | |
| Age, years (Q1-Q3) | 62 (52-73) | 64 (52-69) | 0.604 | 62 (44-72) | 0.495 | 67 (56-74) | 0.086 | 67 (57-72) | 0.264 | 69 (62-76) | **0.020** | 64 (53-73) |
| Male, sex, % (n) | 77.4 (524) | 62.2 (28) | **0.028** | 80.0 (20) | 1.000 | 31.3 (21) | 0.129 | 86.7 (39) | 0.193 | 73.2 (30) | 0.566 | 123 (75.5%) |
| BMI, kg/m² (Q1-Q3) | 26.8 (23.7-30.7) | 27.4 (23.7-30.0) | 0.768 | 26.6 (22.0-28.7) | 0.268 | 26.4 (22.8-30.5) | 0.363 | 26.1 (23.4-30.0) | 0.696 | 24.0 (21.4-29.0) | **0.002** | 26.0 (22.9-30.8) |
| Systolic BP, mmHg (Q1-Q3) | 126 (110-140) | 131 (118-145) | 0.101 | 90 (85-90) | **<0.0001** | 130 (110-150) | 0.598 | 126 (118-135) | 0.693 | 119 (90-140) | 0.005 | 120 (110-130) |
| Diastolic BP, mmHg (Q1-Q3) | 80 (70-88) | 80 (72-82) | 0.523 | 60 (60-67) | **<0.0001** | 75 (70-90) | 0.376 | 80 (70-80) | 0.477 | 70 (60-80) | **<0.0001** | 80 (71-85) |
| Heart rate, bpm (Q1-Q3) | 73 (65-83) | 51 (48-54) | **<0.0001** | 73 (62-84) | 0.546 | 73 (70-90) | 0.760 | 70 (65-86) | 0.548 | 66 (54-85) | **0.011** | 79 (74-88) |
| NYHA class, % (n=887) | | | 0.847 | | 0.072 | | **<0.0001** | | 0.750 | | 0.348 | |
| I | 13.6 (92) | 13.3 (6) | | 0 (0) | | 1.5 (1) | | 8.9 (4) | | 12.2 (5) | | 20 (12.5%) |
| II | 45.5 (308) | 53.3 (24) | | 40.0 (10) | | 31.3 (21) | | 53.3 (24) | | 34.1 (14) | | 88 (55.0%) |
| III | 38.1 (258) | 33.3 (15) | | 60.0 (15) | | 59.7 (40) | | 37.8 (17) | | 48.8 (20) | | 48 (30.0%) |
| IV | 1.5 (10) | 0 (0) | | 0 (0) | | 3.0 (2) | | 0 (0) | | 2.4 (1) | | 4 (2.5%) |
| *Comorbidities* | | | | | | | | | | | | |
| Coronary artery disease, % (n) | 44.3 (300) | 40.0 (18) | 0.643 | 60.0 (15) | 0.152 | 50.7 (33) | 0.443 | 46.7 (21) | 0.759 | 53.7 (22) | 0.261 | 409 (45.4%) |
| Atrial fibrillation, % (n) | 21.3 (144) | 4.4 (2) | **0.004** | 12.0 (3) | 0.326 | 26.9 (18) | 0.281 | 82.2 (37) | 0.707 | 17.1 (7) | 0.693 | 182 (20.2%) |
| Diabetes mellitus, % (n) | 33.1 (224) | 22.2 (10) | 0.134 | 24.0 (6) | 0.465 | 52.2 (35) | **0.004** | 53.3 (24) | **0.003** | 41.5 (17) | 0.542 | 316 (35.1%) |
| Any malignant disease, % (n) | 13.0 (88) | 8.9 (4) | 0.331 | 8.0 (2) | 0.601 | 17.9 (12) | 0.471 | 13.3 (6) | 0.680 | 12.2 (5) | 1.000 | 117 (13.0%) |
| *Medication and Device therapy* | | | | | | | | | | | | |
| Beta-Blocker, % (n) | 88.3 (598) | 93.3 (42) | 0.465 | 76.0 (19) | 0.107 | 86.6 (58) | 0.691 | 93.3 (42) | 0.465 | 92.7 (38) | 0.611 | 88.6 (797) |
| RASi, % (n) | 86.4 (585) | 95.6 (43) | 0.106 | 80.0 (20) | 0.372 | 70.1 (47) | **0.001** | 88.9 (40) | 0.822 | 80.5 (33) | 0.349 | 85.3% (768) |
| MRA, % (n) | 70.6 (478) | 75.6 (34) | 0.611 | 64.0 (16) | 0.506 | 41.8 (28) | **<0.0001** | 66.7 (30) | 0.614 | 61.0 (25) | 0.219 | 67.9% (611) |
| SGLT2i, % (n) | 16.4 (111) | 17.8 (8) | 0.835 | 24.0 (6) | 0.285 | 6.0 (4) | **0.021** | 15.6 (7) | 1.000 | 22.0 (9) | 0.387 | 16.1% (145) |
| Ivabradine, % (n) | 5.5 (37) | 8.9 (4) | 0.314 | 8.0 (2) | 0.644 | 4.5 (3) | 1.000 | 2.2 (1) | 0.504 | 2.4 (1) | 0.717 | 48 (5.3%) |
| Potassium Exchanger, % (n) | 0 (0) | 0 (0) | – | 0 (0) | – | 0 (0) | – | 4.4 (2) | **0.004** | 7.3 (3) | **<0.0001** | 5 (0.6%) |

*(Continued)*

**Table 1.** (Continued)

| | No factors limiting up-titration=677 | HR-group n=45 | p-value | SBP - group n=25 | p-value | eGFR - group n=67 | p-value | HK - group n=45 | p-value | multiple factors limiting up-titration n=41 | p-value | Total cohort n=900 |
|---|---|---|---|---|---|---|---|---|---|---|---|---|
| Diuretics, % (n) | 45.6 (309) | 42.2 (19) | 0.758 | 56.0 (14) | 0.316 | 64.2 (43) | **0.005** | 48.9 (22) | 0.758 | 75.6 (31) | **<0.0001** | 438 (48.7%) |
| ICD, % (n) | 35.2 (238) | 24.4 (11) | 0.065 | 48.0 (12) | 0.315 | 34.3 (23) | 1.000 | 40.0 (18) | 0.758 | 48.8 (20) | 0.233 | 322 (35.8%) |
| CRT, % (n) | 21.6 (146) | 4.4 (2) | **0.003** | 28.0 (7) | 0.190 | 26.9 (18) | 0.484 | 26.7 (12) | 0.373 | 26.8 (11) | 0.681 | 196 (21.8%) |
| PM, % (n) | 10.2 (69) | 4.4 (2) | 0.272 | 4.0 (1) | 0.522 | 7.5 (5) | 0.826 | 8.9 (4) | 0.956 | 7.3 (3) | 0.781 | 84 (9.3%) |
| *Laboratory values* | | | | | | | | | | | | |
| NT-proBNP, pg/ml (Q1-Q3) | 1819.0 (768.4-3831.0) | 949.4 (389.3-1658.5) | **<0.0001** | 3224.0 (1929.0-6027.0) | **0.006** | 6865.5 (3392.0-14531.3) | **<0.0001** | 1947.0 (1015.5-3515.5) | 0.640 | 3259.0 (1295.5-9959.5) | **0.005** | 1951.0 (813.9-4327.5) |
| CREA, mg/dl (Q1-Q3) | 1.1 (0.9-1.3) | 1.1 (0.9-1.2) | 0.791 | 1.4 (1.0-1.7) | **0.011** | 2.9 (2.4-3.7) | **<0.0001** | 1.3 (1.0-1.6) | **0.001** | 2.6 (2.1-4.2) | **<0.0001** | 1.2 (1.0-1.6) |
| eGFR, ml/min/m² (Q1-I3) | 65.6 (49.6-80.1) | 68.0 (46.9-81.9) | 0.813 | 48.3 (43.2-68.9) | 0.017 | 21.4 (14.6-25.9) | **<0.0001** | 52.0 (44.9-69.7) | **0.003** | 22.5 (13.4-29.1) | **<0.0001** | 60.9 (43.4-77.4) |
| Sodium, mmol/l (Q1-Q3) | 140 (138-142) | 140 (138-142) | 0.912 | 139 (135-141) | 0.122 | 139 (137-142) | 0.174 | 138 (137-140) | **0.001** | 138 (136-141) | **<0.0001** | 140.0 (138-142.0) |
| Potassium, mmol/l (Q1-Q3) | 4.7(4.4-4.9) | 4.7 (4.4-5.1) | 0.472 | 4.5 (4.1-4.9) | 0.056 | 4.9 (4.4-5.2) | **0.048** | 5.7 (5.6-6.0) | **<0.0001** | 5.6 (5.0-6.0) | **<0.0001** | 4.7 (4.4-5.1) |
| BChE, U/l (Q1-Q3) | 7.1 (5.6-8.4) | 7.0 (6.0-8.5) | 0.583 | 5.8 (3.8-7.2) | **0.001** | 5.8 (4.4-7.2) | **<0.0001** | 7.1 (5.8-8.7) | 0.870 | 6.0 (4.0-7.1) | **<0.0001** | 6.9 (5.5-8.3) |
| AST (GOT), U/l (Q1-Q3) | 24 (20-30) | 26 (22-31) | 0.214 | 24 (20-35) | 0.672 | 22 (16-29) | **0.037** | 25 (19-34) | 0.858 | 24 (18.3-37.8) | 0.875 | 24.0 (20.0-30.0) |
| ALT (GPT), U/l (Q1-Q3) | 24 (18-36) | 28 (18-38) | 0.490 | 24 (19-39) | 0.826 | 19 (15-28) | **0.002** | 23 (17-33) | 0.686 | 21 (15-29) | 0.018 | 24.0 (17.0-35.0) |
| GGT, U/l (Q1-Q3) | 44 (26-87) | 39 (22-88) | 0.560 | 61 (26-97) | 0.360 | 56 (33-169) | **0.027** | 59 (36-111) | **0.042** | 64 (28-92) | 0.244 | 47.5 (26.0-95.0) |
| Ferritin, µg/L (Q1-Q3) | 158.5 (75.8-253.9) | 200.4 (118.6-291.9) | 0.154 | 138.6 (86.5-177.4) | 0.352 | 204.6 (80.9-198.1) | 0.105 | 159.9 (94.3-331.7) | 0.470 | 189.9 (111.8-396.7) | 0.072 | 162.1 (81.5-263.1) |
| Transferrin saturation, % (Q1-Q3) | 21.0 (14.3-30.0) | 27.6 (21.0-35.7) | **0.005** | 19.4 (11.2-25.8) | 0.255 | 16.6 (11.8-22.9) | **0.002** | 20.1 (14.7-31.5) | 0.953 | 19.4 (13.4-29.3) | 0.544 | 20.7 (14.2-29.8) |
| Leukocyte count, G/l (Q1-Q3) | 7.7 (6.4-9.2) | 7.3 (5.9-8.6) | 0.096 | 8.8 (6.1-10.7) | 0.271 | 7.2 (6.0-9.4) | 0.238 | 8.3 (7.0-10.0) | **0.024** | 7.9 (6.0-9.3) | 0.936 | 7.9 (6.0-9.4) |
| CRP, mg/dl (Q1-Q3) | 0.3 (0.1-0.7) | 0.3 (0.1-0.7) | 0.666 | 0.5 (0.1-2.1) | 0.214 | 0.7 (0.3-2.0) | **<0.0001** | 0.3 (0.1-0.7) | 0.978 | 0.4 (0.2-1.4) | **0.012** | 0.3 (0.1-0.8) |
| Triglycerides, mg/dL (Q1-Q3) | 114.0 (82.0-158.0) | 107.0 (87.8-153.3) | 0.888 | 106.5 (77.5-154.0) | 0.435 | 112.0 (82.5-162.5) | 0.823 | 130.0 (91.0-193.0) | 0.159 | 101.0 (70.0-151.0) | 0.230 | 113.0 (83.0-160.3) |
| Cholesterol, mg/dL (Q1-Q3) | 158.0 (126.0-193.0) | 169.5 (138.5-201.3) | 0.434 | 145.0 (90.5-164.5) | **0.010** | 135.0 (112.0-173.5) | **0.005** | 162.0 (132.5-211.0) | 0.317 | 141.0 (110.5-175.5) | **0.020** | 155.0 (125.0-189.0) |

*(Continued)*

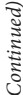

**Table 1.** (Continued)

| | No factors limiting up-titration=677 | HR-group n=45 | p-value | SBP-group n=25 | p-value | eGFR-group n=67 | p-value | HK-group n=45 | p-value | multiple factors limiting up-titration n=41 | p-value | Total cohort n=900 |
|---|---|---|---|---|---|---|---|---|---|---|---|---|
| HDL cholesterol, mg/dL (Q1-Q3) | 47.0 (38.0-57.0) | 45.5 (35.5-54.8) | 0.727 | 42.0 (35.0-52.0) | 0.245 | 45.0 (36.0-57.0) | 0.503 | 46.0 (37.0-55.5) | 0.744 | 45.0 (37.0-56.5) | 0.696 | 46.0 (38.0-57.0) |
| LDL cholesterol mg/dL (Q1-Q3) | 82.2 (57.3-113.3) | 90.2 (66.0-114.2) | 0.380 | 64.4 (46.2-95.6) | **0.023** | 70.6 (44.7-95.6) | **0.005** | 84.8 (61.8-137.0) | 0.321 | 68.6 (42.7-91.7) | **0.010** | 80.2 (55.8-111.2) |
| HbA1c, mg/dL (Q1-Q3) | 6.0 (5.6-6.6) | 5.8 (5.5-6.4) | 0.253 | 5.7 (5.6-6.1) | 0.229 | 6.2 (5.5-6.9) | 0.433 | 6.3 (5.6-7.0) | 0.114 | 6.0 (5.5-6.8) | 0.923 | 6.0 (5.6-6.6) |

*HR – Heart Rate; BP – Blood Pressure; eGFR – estimated Glomerular Filtration Rate; K – Potassium; BMI – Body Mass Index; NYHA – New York Heart Association; RASi – Renin Angiotensin System Inhibition; MRA – Mineralocorticoid Receptor Antagonist; SGLT2i – Sodium Glucose Linked Transporter 2 Inhibition; ICD – Implantable Cardioverter Defibrillator; CRT – Cardiac Resynchronization Rherapy; PM – Pace Maker; NT-proBNP – N-terminal-pro-Brain Natriuretic Peptide; CREA – creatinine BChE – Butyrylcholinesterase; AST - Aspartate aminotransferase; ALT - Alanine aminotransferase; GGT - Gamma-glutamyl transferase; CRP – C-reactive Protein, HDL – high density lipoprotein; LDL – low density lipoprotein; HbA1c – hemoglobin A1c.*

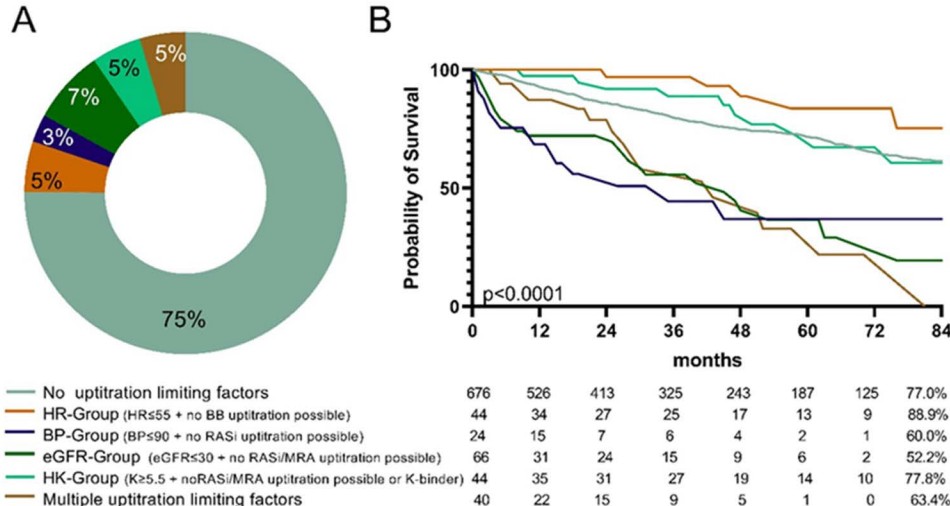

**Fig 1. Distribution and outcome of GDMT up-titration limiting factors. A)** The distribution of factors limiting GDMT up-titration in our cohort is shown as a donut chart. **B)** Kaplan-Meier plots for individual categories show the association between the GDMT up-titration limiting factors and all-cause mortality. For the Kaplan-Meier plot, the difference between groups was assessed using the log-rank test. The p-value is indicated.

The median age was 64 years (Q1-Q3: 53–73), 76.3% of patients were male, and the median value of NT-proBNP was 1951pg/ml (Q1-Q3: 814–4328). At first presentation, 85.3%, 88.6%, and 67.9% of patients received RASi, BB, and MRA, whereas 56.3%, 63.2%, and 66.1% were receiving ≥ 50% of target dosages, respectively.

## Eligibility for GDMT up-titration

At baseline, 677 of 900 patients (75.2%) were clearly eligible for implementation and up-titration of GDMT by showing no factor limiting up-titration, while 24.8% showed at least one up-titration limiting factor as defined by convention. The distribution of up-titration limiting clinical factors is shown in Fig 1 A.

20.2% of patients had one, while 3.4% and 1.1% had two or three up-titration limiting factors, respectively. A comparison between patient groups is shown in Table 1. Impaired renal function with an eGFR ≤ 30 mL/min/1.73 m$^2$ was the most common factor (7.4%), followed by a low RHR (5.0%), presence of HK (5.0%), and a low systolic BP ≤ 90mmHg (2.8%). Patients limited by a RHR of ≤ 55 bpm while not on BB therapy at target dosages showed the lowest risk (NYHA-class: III/IV:33.4%, NT-proBNP 949 pg/ml (Q1-Q3:389–1659; p < 0.0001 comparison for both between all groups), while patients limited by an eGFR ≤ 30ml/min/1.73m2 while not on RASi/MRA therapy at target dosages were characterized by most advanced HF (NYHA-class: III/IV: 62.7%; NT-proBNP 6866 pg/ml (Q1-Q3:3392–14531); p < 0.0001 comparison for both between all groups).

Patients showed a wide range in overall survival rates (p < 0.0001 across all). Patients with bradycardia or with HK demonstrated comparable survival to patients without GDMT up-titration limiting factors with a 5-year survival estimate of 78.9%, 68.0%, and 71.5%, respectively (bradycardia vs. none: p = 0.084; HK vs. none: p = 0.648). Conversely, patients with impaired renal function, low systolic BP, or multiple limitations exhibited a worse survival with a 5-year estimate of 30.4%, 50,4%, and 14.7%, respectively (<0.0001 compared to the other 3 groups).

## GDMT up-titration success

The implementation of drug classes, the up-titration of GDMT, and the outcome of patients respective to up-titration success is shown in Fig 2.

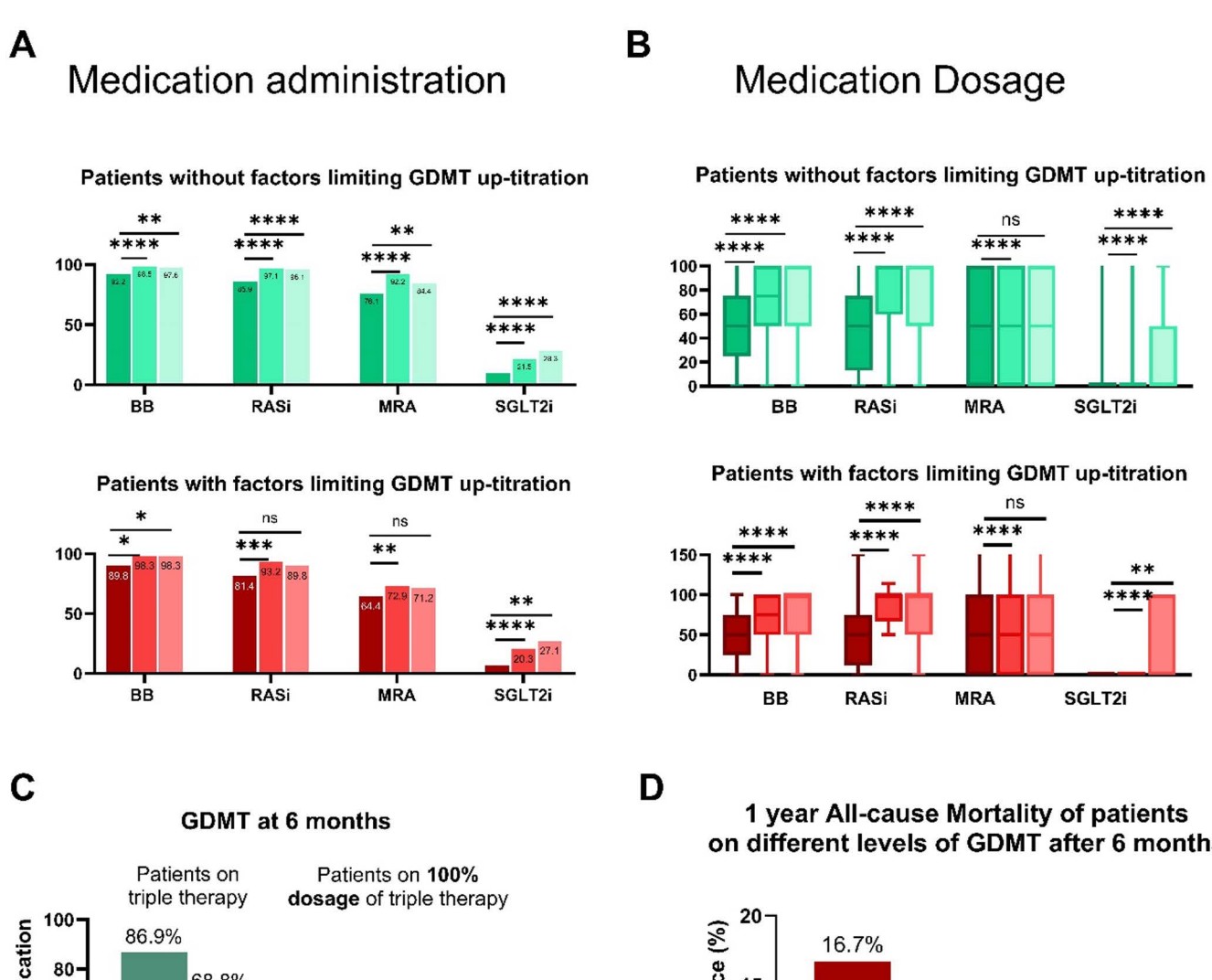

**Fig 2. Use and dosage of HF therapy at baseline, 6 months, and 1 year depending on GDMT up-titration limiting factors. (A) Use and (B) the percent of target dosage** are shown as histograms and Tukey box plots grouped for the respective substance, i.e., BB, RASi, and MRA. The matched Wilcoxon was used for comparisons between baseline vs. 6 months and baseline vs. 1 year. The p-value is indicated as an asterisk in the respective plots. **C** shows the percentage of patients on triple therapy after 6 months, according to the presence of factors limiting GDMT up-titration at baseline (green: no factors limiting GDMT up-titration, red: patients experiencing factors limiting GDMT up-titration). **D** shows 1-year all-cause mortality of patients on different levels of GDMT at 6 months.

There was a significant increase in both medication use and dosage, especially during the first 6 months, regardless of the presence of up-titration limiting factors [without GDMT up-titration limiting factors: p < 0.001 for use, p < 0.001 for dosage and with any GDMT up-titration limiting factor: p ≤ 0.035 for use; p ≤ 0.006 for dosage; comparison between all medication groups]. GDMT up-titration success in detail is shown concerning the up-titration limiting factor in S1 Fig in S1 File.

In summary, the administration of triple therapy (RASi, BB, MRA) at 6 months was achievable in 82,6%, 86.9% and 68.8% of patients with and without any GDMT up-titration limiting factor as a baseline, respectively (Fig 2C). Triple therapy at target dosages (100% of RASi, BB, MRA, according to the current guidelines[2]) could be reached in 35,2% (39.8% in patients without GDMT up-titration limiting factors and 20.3% patients with GDMT up-titration limiting factors) (Fig 2C). Patients with triple therapy and target dosages of 100% exhibited a remarkably good prognosis with a 1-year mortality rate of 2.6%, while patients not reaching the target dosage in any kind of medication had a higher mortality rate of 16.7% (Fig 2D).

## Changes in eligibility for GDMT optimization during up-titration

The trajectories of patients exhibiting up-titration limiting factors over time are shown in Fig 3.

Exact numbers for the trajectories over time are indicated in Table 2.

Counts are given as numbers and percentages; comparison between BL and FUP data was assessed by the Friedman test, and a p-value ≤ 0.05 was considered significant.

There was a notable transition between groups with and without GDMT up-titration limiting factors (p = 0.001 for both baseline vs. 6 months and baseline vs. 1 year). The overall transition between all groups was balanced, especially the proportion of patients without GDMT up-titration limiting factors remained constant at 6 months and 1 year despite optimization of therapy (p = 0.935, Friedman-test).

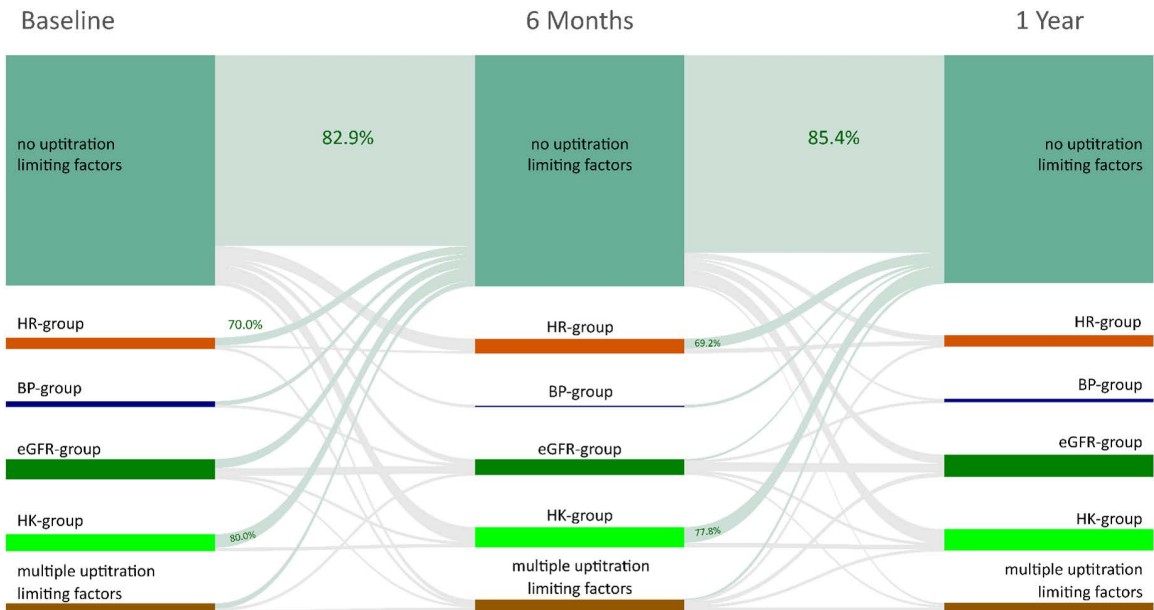

**Fig 3. Factors limiting GDMT up-titration of HF drugs in HFrEF.** The longitudinal evolution of GDMT up-titration limiting factors at baseline, 6-months, and 1-year follow-up (n = 264) is presented as a Sankey diagram.

**Table 2. Trajectories of GDMT up-titration limiting factors over a period of 1 year during up-titration.**

| Up-titration limiting factors Definition: | BL (n=264) | 6 months (n=264) | 1 year (n=264) | p-value |
|---|---|---|---|---|
| None | 77.7% (205) | 78.0% (206) | 76.9% (203) | 0.927 |
| HR ≤ 55 bpm + BB therapy at submaximal dosages | 3.8% (10) | 4.9% (13) | 3.8% (10) | 0.687 |
| systolic BP ≤ 90 mmHg + RASi therapy at submaximal dosages | 1.9% (5) | 0.4% (1) | 1.1% (3) | 0.223 |
| eGFR ≤ 30 ml/min/1.73m2 + RASi and or MRA therapy at submaximal dosages | 6.8% (18) | 5.3% (14) | 7.6% (20) | 0.355 |
| HK (K ≥ 5.5) +RASi and or MRA therapy at submaximal dosages or potassium binder | 5.7% (15) | 6.8% (18) | 7.2% (19) | 0.728 |
| Multiple | 4.2% (11) | 4.5% (12) | 3.4% (9) | 0.692 |

*BL – Baseline; FUP – Follow UP; HR – heart rate; BP – blood pressure; eGFR – estimated glomerular filtration rate; K – potassium; bpm – beats per minute; BB – Beta-Blocker; RASi – Renin-Angiotensin-System-Inhibition; MRA – Mineralocorticoid Receptor Antagonist.*

## The HFA concept: applicability, distribution of clinical profiles and impact on GDMT up-titration

Fig 4 depicts the distribution of the HFA profiles of the study cohort (The mathematical number of all combinations of HFA clinical feature categories results in 108 possible profiles).

Only a third of patients (n=340; 37.8%) could be assigned to a distinct HFA profile. With 560 (62.2%), most patients were non-classifiable (belonging to one of the remaining 97 possible profiles), typically due to the presence of more than one distinguishing clinical feature. Almost all classifiable patients (97.9%) belonged to 4 profiles, i.e., 3, 4, 5, and 7. The other 5 profiles, i.e., 1, 2, 6, 8, and 9, comprised only 7 (2.1%) patients, whereas no patients fulfilled the criteria for profiles 6 and 8. The profiles of non-classifiable patients are summarized in S2 Fig in S1 File.

Fig 4B illustrates the prognosis of HF patients according to the HFA profiles. The 5-year survival estimate differed between the most common profiles: 3, 4, 5, and 7 in descending order, with 89.2%, 81.4%, 68.9%, and 64.4%,

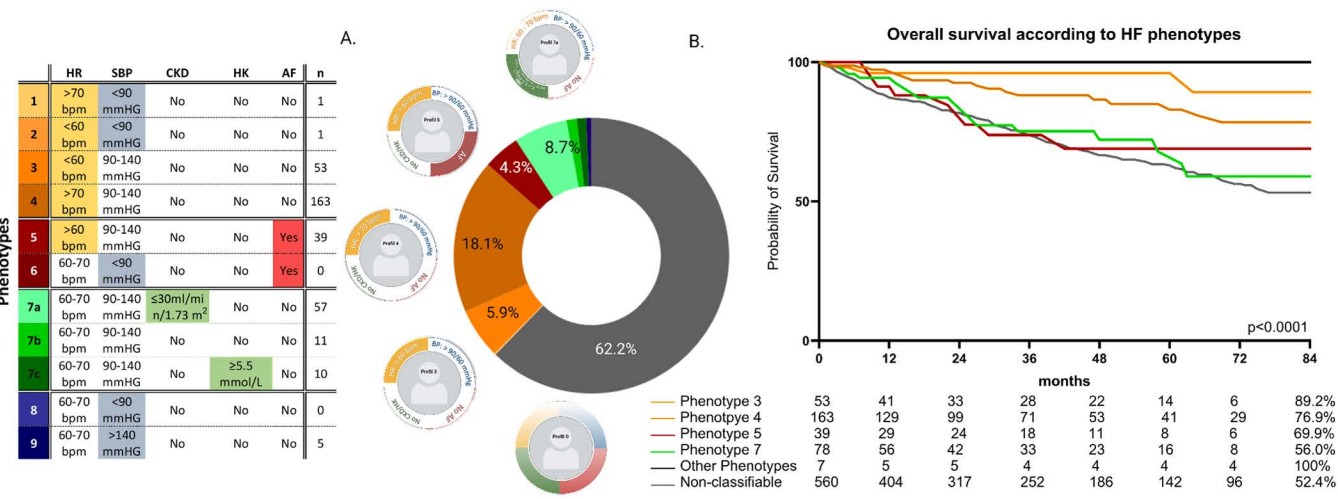

**Fig 4. Phenotype characterization, distribution, and survival. A.** The distribution of all HFA profiles is shown in a donut chart, and 5 adjustments (SBP, 3xCKD, HK (S1 Table in S1 File)) were made. **B**. The association of the HFA profiles with all-cause mortality is shown as Kaplan-Meier plots for individual HFA profiles. For the Kaplan Meier plot, the difference between groups was assessed using the log-rank test. The p-value is indicated in the specific plot.

respectively (p = 0.002 across these profiles). Non-classifiable patients exhibited the worst survival with a 5-year estimate of 62.0% (p < 0.0001 vs. all phenotypes).

The use and dosages of HF-specific medications at baseline, 6-months, and 1-year according to HFA profiles are shown in S3 Fig in S1 File. HF medications significantly increased in use and dosage over time across all main profiles, i.e., 3, 4, 5, and 7 (p ≤ 0.007 for all: BB use/high dosage, RASi use/dosage and MRA use/dosage between baseline and 6-months and baseline and 1 year) and were largely comparable at 6 months and 1 year.

## Discussion

The development of drug therapy for HF has been highly successful over the past three decades. However, the implementation of GDMT at target dosages is incomplete.[2] As one of the main reasons for up-titration failure, physicians claim the presence of supposed up-titration limiting factors. Respective cut-points are not well-defined. To our knowledge, this is the first study to investigate the presence of putative GDMT up-titration limiting factors and their impact on establishing optimal medical therapy in a real-world setting in chronic severe HFrEF patients. The data show that i) with 75%, the majority of patients are unequivocally eligible for the initiation and up-titration of GDMT; ii) both, the medication use and dosage can be successfully improved in patients with and without up-titration limiting factors; iii) the development of up-titration limiting factors during consequent up-titration are rather uncommon and not strictly tied to an individual patient's journey; and iv) patients on optimal medical therapy exhibit a remarkably good survival. Given these results, v) the HFA profiles need to be adapted, as all patients defined by the HFA profiles can be successfully up-titrated while the most severe patients are missed by the classification and, thereby require specific treatment recommendations.

### Under-presciption of GMDT drugs in HFrEF

The development of drug therapy for HF has been highly successful over the past three decades [2]. Currently, four drug classes are recommended as class I, and the establishment as well as the up-titration of GDMT effectively reduces mortality and morbidity in HF [2]. However, despite these recommendations, most patients either do not receive all drug classes or are not at target dosages, as evidenced by large registries [1,3,4,9,19]. This has raised the question about the underlying causes. Currently, three hypotheses are being discussed: firstly, physicians may have limited awareness about HF, resulting in low prescription rates [20,21]; secondly, physicians might exhibit over-caution due to concerns about potential side effects associated with the prescribed medication [22]; and thirdly, patients might not tolerate certain drugs and presence of up-titration limiting clinical factors hinders the successful up-titration of the recommended drugs [23–27]. The results of the recently published clinical studies including real-world cohorts have shown that around 70–80% of patients are primarily eligible for up-titration of all GDMT classes. The cut-offs for factors that supposedly limit up-titration are arbitrary and vary in these studies [11–13].

### Chronic severe HFrEF and GDMT eligibility

In line with these data, in chronic severe HFrEF 75% of patients in our dataset were eligible for GDMT up-titration at baseline. The current study expands this evidence by showing that, most importantly, the same proportion of patients remain eligible despite consequent up-titration of GDMT. Thus, the up-titration of medical therapy is not a question of predisposing factors but, in the majority of HF patients, depends on the treating physician's dedication and patients' compliance. This finding is of particular significance as the data were obtained from a tertiary care center, where patients are typically referred in case they are at an advanced state of HF or if the treating physician deems therapy initiation or up-titration unfeasible due to other reasons. However, the comparability of our cohort with the populations studied in the CHAMP-HF{Greene, 2022 #37} and SWEDE-HF{D'Amario, 2022 #10} registries suggests that our findings are generalizable to the broader HFrEF population, despite being derived from a tertiary care center. During the process of GDMT up-titration, patients may switch between up-titration limiting factors. However, this phenomenon is very balanced. Within

those patients initially not exhibiting any up-titration limitation factor, less than 20% develop a clinical feature potentially limiting further up-titration, yet of those patients who initially presented with any up-titration limiting factor, the same proportion transit to having none at follow-up.

## Clinical factors potentially limiting GDMT up-titration

On the other hand, 25% of patients exhibited clinical factors that warranted caution during the GDMT up-titration. It is important to note that bradycardia, arterial hypotension, renal dysfunction, or HK are not considered absolute contraindications for treatment [2]. The guidelines' authors explicitly state that in such situations, patients should be referred to a specialist, emphasizing the need for further attempts at therapy initiation or up-titration, even in these circumstances [2]. As those patients are usually excluded from clinical studies, there is limited evidence of drug use in these challenging areas [28]. There is, however, also lack of data indicating harm to these patients. This is particularly true for patients with bradycardia or arterial hypotension, which occurs in 9.6% (bradycardia) and 0.8% (hypotension) of outpatients in this study. Regarding renal dysfunction, there is now substantial evidence that the benefits of RASi treatment is still preserved in patients with low eGFR [29], and prospective trials have demonstrated that renal function is not negatively impacted by these drugs [30]. A favorable net balance of missing out on MRA therapy in case of hyperkaliemia is questionable [24,31]. In addition, effective potassium binders are now available and recommended by the current guidelines [2,32].

## Alleged treatment side effects and HF progression

Another crucial point is that all four clinical features described as side effects of therapy (bradycardia, arterial hypotension, renal dysfunction, and HK) also signify HF progression [33]. Therefore, it becomes challenging to differentiate between disease progression and treatment side effects. A recently published paper that summarized the results of large randomized controlled trials (RCTs) demonstrated that arterial hypotension, renal dysfunction, and HK occurring during the studies are primarily caused by the disease itself as compared to the total number of events only to a minority specific to the active treatment arm [33]. Therefore, in our unit physicians made every effort to further increase the administration and dosage of class I medications. Patients already received significant medications at baseline, even if they had one or more of the mentioned factors that warranted caution.

**Transition of up-titration limiting factors.** There was a substantial transition of patients between exhibiting GDMT up-titration limiting factors over time. However, this transition was balanced, with an equal number of patients experiencing improvement and worsening. This might be the result of the combination of effects of both worsening and improvement of HF and side effects of therapy simultaneously. As shown by our data, the net effect of reaching the recommended dosage is beneficial (Fig 2D). As we still lack a predictive tool for individual patient trajectories, every effort must be made to achieve target medication and dosages.

**Prognosis and treatment tolerability.** The question of whether patients with poor prognostic signs, such as hypotension or renal dysfunction, have such high mortality due to the disease severity or due to inadequate drug tolerability is commonly raised. All patient groups received comparable therapy within six months. However, a significant difference remains in the outcomes. Our data suggest that even with similar treatments, prognosis continues to differ. It should be noted that our data cannot definitively answer how much treatment can still modify outcomes in different phenotypes. Larger cohorts of these specific patients, as already shown in the case of RASi [34], need to be investigated to determine treatment effects.

**HFA classification system.** In response, the HFA position paper [16] proposed a classification system whereby patients are categorized into phenotypes based on distinct clinical features that likely impact therapy implementation. Low HR, low BP, impaired renal function and/or hyperkalemia were defined as therapy modulators. The phenotypes come along with corresponding therapy concepts aiming to maximize individual benefit and avoid potential harm. To date, the applicability and distribution of the HFA phenotypes have been analyzed in two cohorts [17,18]. Out of 4455 patients of the

CHECK-HF registry, only 36.8% could be classified into the HFA phenotypes, whereas HFA phenotypes 4, 5, and 7 were the most common, with 11%, 8%, and 6% occurrence, respectively [17]. Musella and colleagues [18] have adapted the clinical strata from the HFA phenotypes and analyzed the distribution of patients into all possible combinations, resulting in 108 phenotypes. The authors have found that most patients fit into a few easily identifiable categories, e.g., presence of AF with good renal function, no HK and HR > 70bpm (14.3%), AF with eGFR 30–60 and HR > 70bpm (10.7%) and patients without AF eGFR > 60, HR > 70bpm, all with normal systolic BP with 90–140mmHg, to name the three most frequent. In line with these results, only 38% of our cohort could also be classified into the HFA phenotypes. Phenotypes 3, 4, 5, and 7 were the most common. The non-classifiable patient group presents with heterogeneous clinical profiles and prognoses, including very low- and very high-risk patients. This group is worth further discrimination when it comes to the optimization of HF medication and identifying high-risk patients.

Our data suggest that the inclusion and exclusion criteria used in large RCTs should be more liberal to gain further insights into these high-risk patients. By including a broader range of patients with severe prognostic indicators, we can better understand the effects of HF treatment in this patient population.

Limitations: Our study was conducted retrospectively at a single tertiary medical care center, which may limit the generalizability of our findings. Although we achieved a high proportion of patients reaching target dosages for triple therapy, this may not be reflective of broader clinical practice. The intensive management and close monitoring provided in our center may not be feasible in all settings, potentially limiting the reproducibility of our results.

## Conclusion

This real-world study underlines the feasibility of up-titration of HF treatment in significantly more patients than might be presumed based on registries. Confirming other data, even in chronic severe heart failure 75% of real-world HFrEF patients are eligible for GDMT up-titration at the initial presentation and throughout the up-titration process. In 25% of patients, there is a considerable transition between GDMT up-titration limiting factors, which do not seem to be tied to an individual patient´s journey, but appear dynamic and evolving, highlighting the need for continuous reassessment.

## Supporting information

**S1 File. Additional data and methods supporting the findings of this study are available in the Supporting information. Supplementary Figure 1. Use and dosage of HF therapy at baseline, 6 months, and 1 year depending on GDMT up-titration limiting factors. Supplementary Figure 2 Non-classifiable patients: characterization, distribution and survival. Supplementary Figure 3. Use and dosage of HF therapy at baseline, 6 months and 1 year depending on GDMT up-titration limiting factors. Supplementary Table 1. Modified phenotypes based on the classification by Rosano et al. Supplementary Methods 1. The HFA concept classification of the HF clinical profiles.**
(DOCX)

## Author contributions

**Conceptualization:** Annika Weidenhammer, Suriya Prausmüller, Noel Panagiotides, Martin Hülsmann, Noemi Pavo.

**Data curation:** Annika Weidenhammer, Suriya Prausmüller, Marc Stadler, Noel Panagiotides, Georg Spinka, Gregor Heitzinger, Henrike Arfsten, Anush Barkhudaryan, Martin Hülsmann, Noemi Pavo.

**Formal analysis:** Annika Weidenhammer, Suriya Prausmüller, Marc Stadler, Noel Panagiotides, Anush Barkhudaryan, Noemi Pavo.

**Investigation:** Annika Weidenhammer, Anush Barkhudaryan, Clemens Partsch, Martin Hülsmann.

**Methodology:** Annika Weidenhammer, Clemens Partsch, Martin Hülsmann, Noemi Pavo.

**Project administration:** Philipp Bartko, Christian Hengstenberg, Martin Hülsmann.

**Resources:** Annika Weidenhammer, Henrike Arfsten, Clemens Partsch, Philipp Bartko, Georg Goliasch, Christian Hengstenberg, Martin Hülsmann, Noemi Pavo.

**Software:** Annika Weidenhammer, Guido Strunk.

**Supervision:** Georg Spinka, Gregor Heitzinger, Henrike Arfsten, Guido Strunk, Philipp Bartko, Georg Goliasch, Christian Hengstenberg, Martin Hülsmann, Noemi Pavo.

**Validation:** Guido Strunk, Georg Goliasch, Noemi Pavo.

**Visualization:** Annika Weidenhammer, Marc Stadler, Gregor Heitzinger.

**Writing – original draft:** Annika Weidenhammer, Martin Hülsmann, Noemi Pavo.

**Writing – review & editing:** Annika Weidenhammer, Martin Hülsmann, Noemi Pavo.

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
