## [Decision Letter · Decision Letter 0]

2 Mar 2025

PONE-D-24-54347Eligibility and GDMT up-titration success in heart failure.

A real-world assessmentPLOS ONE

Dear Dr. Hülsmann,

Thank you for submitting your manuscript to PLOS ONE. After careful consideration, we feel that it has merit but does not fully meet PLOS ONE’s publication criteria as it currently stands. Therefore, we invite you to submit a revised version of the manuscript that addresses the points raised during the review process.

We look forward to receiving your revised manuscript.

Kind regards,

Satoshi Higuchi

Academic Editor

PLOS ONE

Journal Requirements:

Reviewers' comments:

Reviewer's Responses to Questions

**Comments to the Author**

1. Is the manuscript technically sound, and do the data support the conclusions?

Reviewer #1: Yes

Reviewer #2: Partly

2. Has the statistical analysis been performed appropriately and rigorously? 

Reviewer #1: Yes

Reviewer #2: Yes

3. Have the authors made all data underlying the findings in their manuscript fully available?

Reviewer #1: Yes

Reviewer #2: Yes

4. Is the manuscript presented in an intelligible fashion and written in standard English?

Reviewer #1: Yes

Reviewer #2: No

5. Review Comments to the Author

Reviewer #1: The study investigates the presence of limiting factors in the gradual titration of GDMT in patients with advanced HFrEF in real clinical settings and its impact on the establishment of optimal pharmacotherapy. It presents very interesting findings.

However, I feel that several revisions are needed for the manuscript, as detailed below.

1. The cohort in this study is derived from a single tertiary medical center. How much do the results differ from those in a more general population of HFrEF patients? To what extent are the findings generalizable?

2. There is no clear mention of the limitations of the study.

3. In the Supplementary Methods (Section 1), it is stated that for AF, resting HF is categorized as 60-70 and >60. Is this correct?

4. Was the actual adjustment of medications based on the cut-off values for the GDMT titration limiting factors as specified in this study, or was it left to the discretion of the attending physicians? For instance, is it possible that RASI medications were up-titrated even when the eGFR was less than 30?

5. In the first paragraph of page 12, it is written: "v) the HFA profiles need to be adapted, as all patients defined by the HFA profiles can be successfully up-titrated while the most severe patients are missed by the classification and, thereby require specific treatment recommendations." In the position paper, however, it states: "Of course, physicians will recognize patients cannot always be characterized accurately by simple demographics, so that advice may need to be sought by comparison and combinations of the advice for one or more profiles." The statement in the manuscript feels somewhat assertive. Can you propose a new classification to identify high-risk patients who are not included in the HFA profiles?

Reviewer #2: Overall, the authors present work detailing real world experiences with GDMT use in patients with HFrEF. The author should be commended for completing this important work, as this problem remains central to continued poor outcomes among those with HFrEF. There are grammatical errors throughout the text that should be fixed. Additionally, there are a few language syntax errors that could be improved to better portray the message of the authors.

-------

Introduction – Please abbreviate GDMT in the first instance. Additionally, it would be valuable to mention HFrEF in the first sentence to set the stage for the rest of the paper, especially given the lack of substantial class I indicated medications for other types of HF.

In the second paragraph, the sentence starting with “Other Data” should be referenced so the reader can refer to the data highlighted by the authors. Additionally, the second sentence on page 5 “This study has used different” is confusing and unclear what point is being made. Was it the Greene study that utilized different cut-offs? Did the study just demonstrate the different cutoffs used in standard practice? Does GWTG data record why people did not receive medications?

Was the aim of the paper just to look at uptitration of GDMT or was it to evaluate the HFA phenotypes impact or association with uptitration?

Methods – What does an “all-comer” clinic mean? Who gets referred there? People after a hospitalization or people from the community? Is the clinic staffed by cardiologists or heart failure trained cardiologists? APPs? Does the definition offered in the patient description actually represent “advanced” HF? A low ejection fraction and elevated BNP just means they have HFrEF, not necessarily “advanced” HF, which is usually represented by persistent symptoms despite adequate therapy.

In the introduction, the authors introduce the strong-HF exclusion criteria as a source of confusion regarding uptitration success but then cite the same paper in their methodology for developing the “limiting factors” component. Maybe in the introduction, they could reword this reference to something along the lines of, “Implementation trials, like STRONG-HF, use established cutoffs for up titration of medication, and these present challenges to providing adequate therapy.” This could help introduce the commonly help “cut-offs” that will ultimately be used for this analysis.

How did the authors determine target doses? Is this from the trial data for each of the medication classes?

Analysis – Were KM curves done for each individual limiting factor, or whether a participant had ANY limiting factor vs. NONE. “GDMT at 6 months was calculated in comparison to all patients with data at 6 months” is unclear and the reader may have a hard time understanding what comparison was done here.

Results – the phrase “clearly eligible” should be defined. Does this mean having no limiting factors?

The HFA profiles/phenotypes/etc language throughout the manuscript is confusing. It would be helpful to refer to these profiles with one common name as to not confuse the reader. In the intro, it is introduced as a 9 profile system, but in the results, there are 108 different profiles? It would be helpful to spend the time explaining this concept in the methods given that it is a major part of the analysis plan introduced by the authors. Additionally, how useful is the criteria when >60% of participants are not defined?

Discussion – It would be helpful for the authors to better clarify why the population studied was “advanced” HF. The criteria presented earlier was not a strong representation of advanced disease, but if there is other evidence that the definition proposed by the authors is reasonable, then it would be helpful to cite.

On page 13 of the discussion, the authors reference the “treating physician’s dedication and patients’ compliance,” as the contributing factors to lack of titration given the broad eligibility of the patients studied. This point could be strengthened if the authors add greater detail to the patient mix. Are these patients post-hospitalization? Are they from ambulatory clinics? Are these new HF patients or do they have prevalent disease?

What does the final sentence mean in the conclusion? Do the other 25% of patients have alternating limiting factors between time points? How are they not tied to the patient journey? Does journey mean worsening outcomes?

6. PLOS authors have the option to publish the peer review history of their article (what does this mean? ). If published, this will include your full peer review and any attached files.

**Do you want your identity to be public for this peer review?** For information about this choice, including consent withdrawal, please see our Privacy Policy .

Reviewer #1: No

Reviewer #2: No

---

## [Decision Letter · Decision Letter 1]

17 Apr 2025

Eligibility and GDMT up-titration success in heart failure.

A real-world assessment

PONE-D-24-54347R1

Dear Dr. Hülsmann,

We’re pleased to inform you that your manuscript has been judged scientifically suitable for publication and will be formally accepted for publication once it meets all outstanding technical requirements.

Kind regards,

Satoshi Higuchi

Academic Editor

PLOS ONE

Additional Editor Comments (optional):

Reviewers' comments:

Reviewer's Responses to Questions

**Comments to the Author**

1. If the authors have adequately addressed your comments raised in a previous round of review and you feel that this manuscript is now acceptable for publication, you may indicate that here to bypass the “Comments to the Author” section, enter your conflict of interest statement in the “Confidential to Editor” section, and submit your "Accept" recommendation.

Reviewer #1: All comments have been addressed

Reviewer #2: All comments have been addressed

2. Is the manuscript technically sound, and do the data support the conclusions?

Reviewer #1: Yes

Reviewer #2: Yes

3. Has the statistical analysis been performed appropriately and rigorously? 

Reviewer #1: Yes

Reviewer #2: Yes

4. Have the authors made all data underlying the findings in their manuscript fully available?

Reviewer #1: Yes

Reviewer #2: Yes

5. Is the manuscript presented in an intelligible fashion and written in standard English?

Reviewer #1: Yes

Reviewer #2: Yes

6. Review Comments to the Author

Reviewer #1: (No Response)

Reviewer #2: Authors addressed and responded to each comment adequately. Specifically, they addressed the major components that could result in confusion regarding their findings. Great work.

7. PLOS authors have the option to publish the peer review history of their article (what does this mean? ). If published, this will include your full peer review and any attached files.

**Do you want your identity to be public for this peer review?** For information about this choice, including consent withdrawal, please see our Privacy Policy .

Reviewer #1: No

Reviewer #2: No

---

## [Editor Report · Acceptance letter]

PONE-D-24-54347R1

PLOS ONE

Dear Dr. Hülsmann,

I'm pleased to inform you that your manuscript has been deemed suitable for publication in PLOS ONE. Congratulations! Your manuscript is now being handed over to our production team.

Kind regards,

on behalf of

Dr. Satoshi Higuchi

Academic Editor

PLOS ONE